# EMERGING SEMANTIC SEGMENTATION FROM POSITIVE AND NEGATIVE COARSE LABEL LEARNING

## ABSTRACT

Large annotated dataset is of crucial importance for developing machine learning models for segmentation. However, the process of producing labels at the pixel level is time-consuming, error-prone, and in medical imaging requires expert annotators, which is also expensive. We note that it is simpler and less costly to draw merely rough annotations, e.g., coarse annotations, which reduce the effort for expert and non-expert level annotators. In this paper, we propose to use coarse drawings from both positive (e.g., objects to be segmented) and negative (objects not to be segmented) classes in the image, even with noisy pixels, to train a convolutional neural network (CNN) for semantic segmentation. We present a method for learning the true segmentation label distributions from purely noisy coarse annotations using two coupled CNNs. The separation of the two CNNs is achieved by high fidelity with the characters of the noisy training annotations. We propose to add a complementary label learning that encourages estimating negative label distribution. To illustrate the properties of our method, we first use a toy segmentation dataset based on MNIST. We then present the quantitative results on publicly available datasets: Cityscapes dataset for multi-class segmentation, and retinal images for medical applications. In all experiments, our method outperforms the state-of-the-art methods, particularly in the cases where the ratio of coarse annotations is small compared to the given dense annotations. *The code will be released after reviewing.*

## 1 INTRODUCTION

Thanks to the availability of large datasets with accurate annotations Deng et al. (2009), Fully Supervised Learning (FSL) has been translated from theoretical algorithms to practice. However, it is often expensive, time-consuming, and perhaps impossible to collect pixel-level labels for large-scale datasets Jing et al. (2019). This problem is particularly prominent in the medical domain where the labeled data are commonly scarce due to the high cost of annotations Zhang et al. (2020b). For instance, accurate segmentation of vessels in fundus retinal images is difficult even for experienced experts due to variability of vessel's location, size, and shape across population or disease (see Fig. S1 in Appendix). The labeling process is prone to errors, almost inevitably leading to noisy datasets as seen in machine learning benchmark datasets Peterson et al. (2019). Labeling errors can occur due to automated label extraction, ambiguities in input and output spaces, or human errors (e.g. lack of expertise). Those errors can be further exacerbated by data and human biases (e.g. level of expertise of the annotator), and thus annotations of structures e.g. organs in medical images, suffer from high inter- and intra- observer variability. As a consequence, despite the availability of large imaging repositories (e.g. UK Biobank Littlejohns et al. (2020)), the generation of the curated labels that are available to machine learning remains a challenging issue, necessitating the development of methods that learn robustly from noisy annotations.

Recently, several foundation models Kirillov et al. (2023); Ma & Wang (2023); Butoi et al. (2023) are proposed for universal image segmentation. However, these models may output poor results or even totally fails for objects with irregular shapes, weak boundaries, small sizes, or low contrast. For most situations, the subpar segmentation performance is not sufficient and satisfying especially for medical image segmentation where extremely high accuracy are demanded. Therefore, curating annotations for specific supervised learning tasks is generally necessary and crucial for achieving accurate and meaningful results. To reduce the workload of pixel-level annotation, there has been a considerable effort to exploit semi-supervised and weakly-supervised strategies. Semi-Supervised Learning (SSL) aims at training networks jointly with a small number of labeled data and a large amount of unlabeled data. Extensive research on SSL shows that plenty of unlabeled data would contribute to the model performance and generalization abilities Chapelle et al. (2009); Cheplygina et al. (2019). However, the additional information from unlabeled data can improve CNNs only under specific assumptions Chapelle et al. (2009), and SSL still requires representative image and annotation pairs to be available. In turn, Weakly-supervised Learning (WSL) utilizes annotations that are cheaper to produce than pixel-wise labels such as bounding boxes Dai et al. (2015); Jiang et al.

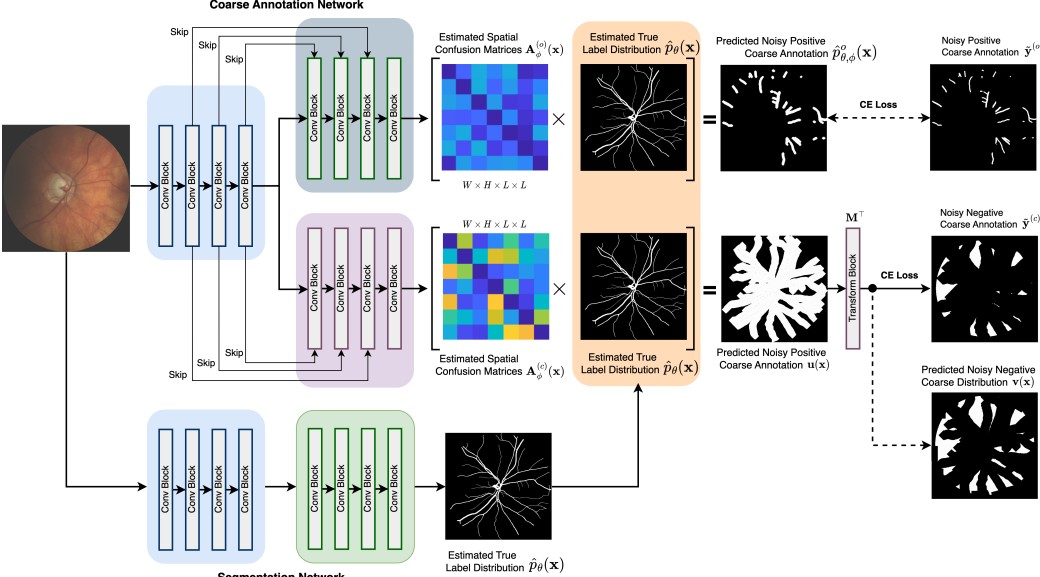

Figure 1: General schematic of the model supervised by noisy coarse annotations. The method consists of two components: (1) Segmentation network parameterized by $\theta$ that generates an estimate of the true segmentation probabilities $\hat{p}_\theta(\mathbf{x})$ for the given input image $\mathbf{x}$; (2) Coarse annotation network, parameterized by $\phi$, that estimates the confusion matrices (CMs) $\{\mathbf{A}_\phi^{(o)}(\mathbf{x}), \mathbf{A}_\phi^{(c)}(\mathbf{x})\}$ of the noisy coarse annotations.

(2022), coarse annotation Wang et al. (2021); Saha et al. (2022), scribbles Lin et al. (2016); Zhang & Zhuang (2022), or image-level labels Papandreou et al. (2015); Pathak et al. (2014) to train the segmentation models. These weak annotations increase the efficiency of the manual labeling procedure and achieve promising performance making the gap between the SSL and FSL performance smaller. However, the information from weak annotations is of lower precision and usually suffers from noisy information when curating the labels, e.g. the uncertainty on the boundary. For example, bounding box indicates the rough positions of the object of interest (OOI), but the pixels inside the bounding box may belong to multiple classes if the box size is large. A more annotator-friendly way to get supervision efficiently is scribble-based annotation, which only requires the annotator to draw a few lines to mark a small part of the OOI. Coarse annotations can provide much more information than scribble and avoid large non-target pixels being grabbed into the bounding box. Meanwhile, drawing coarse annotations on images (e.g., rough boundaries for the OOI and non-OOI) needs only similar effort and time as the scribble and box-level labeling, and can be conducted by non-experts. Therefore, learning from those coarse annotations, and then correcting the noisy pixels with computational methods, might be an efficient way to enrich the labeled large-scale data with minimal effort.

**Our contribution:** We introduce the first instance of an end-to-end supervised segmentation method that estimates true segmentation labels from noisy coarse annotations. The proposed architecture (see Fig. 1) consists of two coupled CNNs where the first CNN estimates the true segmentation probabilities, and the second CNN models the characteristics of two different coarse annotations i.e. positive annotation indicates the region of interest and negative annotation indicates the coarse non-target region e.g. background) by estimating the pixel-wise confusion matrices (CMs) on a per-image basis. Unlike the previous WSL methods using coarse annotations, our method models and disentangles the complex mappings from the input images to the noisy coarse annotations and to the true segmentation label simultaneously. Specifically, we model the noisy coarse annotation for the objective along with the complementary label learning for the background or non-objective to enable our model to disentangle robustly the errors of the given annotations and the true labels, even when the ratio of coarse annotation is small (e.g., given scribble for each class). In contrast, this would not be possible with the other WSL methods where the parameters of each coarse annotation are estimated on every target image separately.

For evaluation, we first simulate the noisy positive and negative coarse annotations on the MNIST dataset enhanced by morphometric operations with Morpho-MNIST framework Castro et al. (2019); then we provide an extensive comparison of our method on the publicly available reference Cityscapes dataset for multi-class segmentation tasks; finally, we show the robustness of our method on challenging medical image segmentation. The presented experiments demonstrate that our method consistently leads to better segmentation performance compared to the state-of-the-art SSL, WSL methods, and other relevant baselines, especially when the ratio of coarse annotation for each class is low and the degree of annotation noise is high.

## 2 RELATED WORK

**Weakly-Supervised Segmentation.** To reduce the labor-intensive, pixel-wise labeling, different forms of weak annotations for segmentation were explored including scribbles Lin et al. (2016); Zhang & Zhuang (2022), bounding box Dai et al. (2015); Jiang et al. (2022), points Bearman et al. (2016), image-level labels Papandreou et al. (2015); Pathak et al. (2014), etc. Lin et al. (2016) proposed SribbleSup to apply graph-based methods to propagate information to the unlabeled pixels. Zhang & Zhuang (2022) proposed CycleMix to learn segmentation from scribbles and enhance the generalization ability of segmentation models. Dai et al. (2015) investigated bounding box annotations as an alternative or extra source of supervision to train neural networks for semantic segmentation. L2G Jiang et al. (2022) took advantage of both the global and the local views randomly cropped from the input image for knowledge transfer on semantic segmentation.

Another strategy to reduce amount of manual annotations is Complementary Label Learning (CLL), which is a WSL framework where each sample is equipped with a complementary label that denotes one of the classes the sample does not belong to Ishida et al. (2017) Gao & Zhang (2021). Ishida et al. (2017) proposed learning from examples with only complementary labels by assuming that a complementary label is uniformly selected from the $L-1$ classes other than the true label class ($L<2$). Gao & Zhang (2021) proposed a risk estimator with guaranteed estimation error bound based on a discriminative model for CLL and further introduced weighted loss to the empirical risk to maximize the predictive gap between potential ground-truth (GT) label and complementary label. Yu et al. (2018) introduced a general method to modify traditional loss functions and extended standard deep neural network classifiers to learn from noisy complementary labels.

As presented above, the segmentation methods focused on the weak annotation either for the target objective or for the complementary label. Contrary, we proposed to exploit both of them simultaneously, as both the target and complementary classes can provide weak supervision to train a better model.

**Learning from Noisy Labels.** In standard classification problems, modeling the label noise statistically has been widely studied e.g. by introducing a probability transition matrix. For example, previous works Natarajan et al. (2013); Liu & Tao (2015); Wang et al. (2017) employ probability transition matrix to modify loss functions such that they can be robust to noisy labels. Furthermore, Patrini et al. (2017); Sukhbaatar et al. (2015) extended such probability transition matrix to a transition layer and implement it with deep neural networks (DNN). More recently, Zhang et al. (2023) utilized a DNN to learn the characteristics of individual annotators by estimating the pixel-wise confusion matrices (CMs) on a per-image basis, which mimic the annotation errors.

However, this is the first time that this idea is applied to the new problem of learning with noisy coarse annotation. More broadly, our work is related to methods for robust learning in the presence of coarse annotations with noisy information (e.g., over- or missing-pixel segmentation). There is a large body of literature Wang et al. (2021); Saha et al. (2022); Zhang & Zhuang (2022) that does not explicitly model noisy coarse annotations, unlike our method.

## 3 METHOD

### 3.1 PROBLEM SET-UP

In this work, we consider developing a supervised segmentation model by learning the positive (e.g. object to be segmented) and negative (e.g. object to be not segmented) coarse annotations that are easy and less expensive to be acquired from annotators. Specifically, we consider a scenario where set of images $\{\mathbf{x}_n \in \mathbb{R}^{W \times H \times C}\}_{n=1}^N$ (with $W,H,C$ denoting the width, height and depths of the image) are assigned with coarse segmentation labels $\{\{\tilde{\mathbf{y}}_n^{(o)}, \tilde{\mathbf{y}}_n^{(c)}\} \in \mathcal{Y}^{W \times H}\}_{n=1,...,N}^{\{o,c\} \in S(\mathbf{x}_i)}$ from objective and complementary categories where $\tilde{\mathbf{y}}_n^{(o)}$ and $\tilde{\mathbf{y}}_n^{(c)}$ denote the noisy objective and complementary coarse annotations, respectively and $S(\mathbf{x}_n)$ denotes the set of all annotations of image $\mathbf{x}_i$ and $\mathcal{Y}=[1,2,...,L]$ denotes the set of segmentation classes.

Our problem can be easily understood as *learning the unobserved true segmentation distribution $p(\mathbf{y}|\mathbf{x})$ from the dataset $\mathcal{D} = \{\mathbf{x}_n, (\tilde{\mathbf{y}}_n^{(o)}, \tilde{\mathbf{y}}_n^{(c)})\}_{n=1,...,N}^{\{o,c\} \in S(\mathbf{x}_n)}$* with coarse drawing labels, i.e., the combination of images, noisy coarse annotations from both positive and negative categories. We assume the model works even for every image $\mathbf{x}$ annotated by only one category of coarse annotations, either the

positive or the negative coarse annotation is provided, i.e., $|S(\mathbf{x})| \geq 1$. During the training, GT labels $\{\mathbf{y}_n \in \mathcal{Y}^{W \times H}\}_{n=1,\dots,N}$ are not available.

## 3.2 PROBABILISTIC MODEL OF NOISY COARSE LABELS

In this section, we outline the probabilistic model used to describe the observed positive and negative coarse annotations. Given the input image, we assume that the provided coarse annotations are generated statistically independently across different samples and over different pixels. Under these assumptions, the probability of observing coarse labels $\{(\tilde{\mathbf{y}}_n^{(o)}, \tilde{\mathbf{y}}_n^{(c)})\}_{\{o,c\} \in S(\mathbf{x})}$ on the image $\mathbf{x}$ factorises as:

$$p(\{\tilde{\mathbf{y}}^{(o)}, \tilde{\mathbf{y}}^{(c)}\}_{\{o,c\} \in S(\mathbf{x})} | \mathbf{x}) = \prod_{\{o,c\} \in S(\mathbf{x})} p(\{\tilde{\mathbf{y}}^{(o)}, \tilde{\mathbf{y}}^{(c)}\} | \mathbf{x}) = \prod_{\substack{\{o,c\} \in S(\mathbf{x})}} \prod_{\substack{w \in \{1,\dots,W\} \\ h \in \{1,\dots,H\}}} p(\{\tilde{y}_{wh}^{(o)}, \tilde{y}_{wh}^{(c)}\} | \mathbf{x}) \quad (1)$$

where $\tilde{y}_{wh}^{(o)}, \tilde{y}_{wh}^{(c)} \in [1,\dots,L]$ denotes the $(w,h)^{\text{th}}$ elements of $\{\tilde{\mathbf{y}}^{(o)}, \tilde{\mathbf{y}}^{(c)}\} \in \mathcal{Y}^{W \times H}$. Now we rewrite the probability of observing coarse label on each pixel $(w,h)$ as:

$$p(\{\tilde{y}_{wh}^{(o)}, \tilde{y}_{wh}^{(c)}\} | \mathbf{x}) = \sum_{y_{wh}=1}^{L} p(\{\tilde{y}_{wh}^{(o)}, \tilde{y}_{wh}^{(c)}\} | y_{wh}, \mathbf{x}) \cdot p(y_{wh} | \mathbf{x}) \quad (2)$$

where $p(y_{wh} | \mathbf{x})$ denotes the GT label distribution over the $(w,h)^{\text{th}}$ pixel in the image $\mathbf{x}$, and $p(\{\tilde{y}_{wh}^{(o)}, \tilde{y}_{wh}^{(c)}\} | y_{wh}, \mathbf{x})$ describes the coarse labeling process by which category of the label belongs to. In particular, we refer to the $L \times L$ matrix whose each $(i,j)^{\text{th}}$ element is defined by the second term $\mathbf{a}^{\{o,c\}}(\mathbf{x},w,h)_{ij} := p(\tilde{y}_{wh}^{\{o,c\}} = i \mid y_{wh} = j, \mathbf{x})$ as the CM of the corresponding coarse label at pixel $(w,h)$ in image $\mathbf{x}$.

Our approach involves a CNN-based architecture that is designed to effectively model the various constituents within the joint probability distribution described above $p(\{\tilde{\mathbf{y}}^{\{o,c\}}\}_{\{o,c\} \in S(\mathbf{x})} | \mathbf{x})$ as illustrated in Fig. 1. The model consists of two components: (1) *Segmentation Network*, parameterized by $\theta$, which estimates the true segmentation probability map, $\hat{\mathbf{p}}_\theta(\mathbf{x}) \in \mathbb{R}^{W \times H \times L}$ whose each $(w,h,i)^{\text{th}}$ element approximates $p(y_{wh} = i \mid \mathbf{x})$; and (2) *Coarse Annotation Network*, parameterized by $\phi$, that estimates the pixel-wise CMs of respective annotations as a function of the input image, $\{\hat{\mathbf{A}}_\phi^{\{o,c\}}(\mathbf{x}) \in [0,1]^{W \times H \times L \times L}\}$ whose each $(w,h,i,j)^{\text{th}}$ element approximates $p(\tilde{y}_{wh}^{\{o,c\}} = i \mid y_{wh} = j, \mathbf{x})$. Each product $\hat{\mathbf{p}}_{\theta,\phi}^{\{o,c\}}(\mathbf{x}) := \hat{\mathbf{A}}_\phi^{\{o,c\}}(\mathbf{x}) \cdot \hat{\mathbf{p}}_\theta(\mathbf{x})$ represents the estimated segmentation probability map of the corresponding category of annotation. Note that here "·" denotes the element-wise matrix multiplications in the spatial dimensions $W, H$. At inference time, we use the output of the segmentation network $\hat{\mathbf{p}}_\theta(\mathbf{x})$ to segment the testing images.

## 3.3 LEARNING FROM NOISY COARSE ANNOTATIONS

In this section, we describe how we jointly optimize the parameters of the segmentation network, $\theta$, and the parameters of the coarse annotation network, $\phi$. In short, we minimize the negative log-likelihood of the probabilistic model for both positive and negative coarse annotations via stochastic gradient descent. A detailed description is provided below.

**Learning with Positive Coarse Label.** Given training input $\mathbf{X} = \{\mathbf{x}_n\}_{n=1}^N$ and positive coarse labels $\tilde{\mathbf{Y}}^{\{o\}} = \{\tilde{\mathbf{y}}_n^{\{o\}} : \{o\} \in S(\mathbf{x}_n)\}_{n=1}^N$, we optimize the parameters $\{\theta, \phi\}$ by minimizing the negative log-likelihood (NLL), $-\log p(\tilde{\mathbf{Y}}^{(1)}, \dots, \tilde{\mathbf{Y}}^{(O)} | \mathbf{X})$. From Eq. (1) and Eq. (2), this optimization objective equates to the sum of cross-entropy (CE) losses between the observed positive coarse segmentation and the predicted label distributions:

$$-\log p(\tilde{\mathbf{Y}}^{(1)}, \dots, \tilde{\mathbf{Y}}^{(O)} | \mathbf{X}) = \sum_{n=1}^{N} \sum_{o=1}^{O} \mathbb{1}(o \in \mathcal{S}(\mathbf{x}_n)) \cdot \text{CE}\left(\hat{\mathbf{A}}_\phi^{(o)}(\mathbf{x}_n) \cdot \hat{\mathbf{p}}_\theta(\mathbf{x}_n), \tilde{\mathbf{y}}_n^{(o)}\right) \quad (3)$$

Minimizing Eq. (3) encourages the positive-specific predictions $\hat{\mathbf{p}}_{\theta,\phi}^{(o)}(\mathbf{x})$ to be as close as possible to the provided positive coarse label distributions $\mathbf{p}^{(o)}(\mathbf{x})$. However, this loss function alone cannot separate the annotation noise from the true label distribution; there are many combinations of pairs $\hat{\mathbf{A}}_\phi(\mathbf{x})$ and segmentation model $\hat{\mathbf{p}}_\theta(\mathbf{x})$ such that $\hat{\mathbf{p}}_{\theta,\phi}(\mathbf{x})$ matches well the provided distribution $\mathbf{p}(\mathbf{x})$ for any input

image $\mathbf{x}$ (e.g. permutations of rows in the CMs). Tanno et al. (2019) addressed an analogous issue for the classification task, and here we add the trace of the estimated CMs to the loss function for positive coarse annotation in Eq.(3) as a regularisation term. We thus optimize the regularized loss:

$$
\begin{aligned}
\mathcal{L}_{\text{obj}}(\theta,\phi) &:= \mathcal{L}_{\text{obj}}(\theta,\phi)(\hat{\mathbf{A}}_{\phi}^{(o)}(\mathbf{x}_n) \cdot \hat{\mathbf{p}}_{\theta}(\mathbf{x}_n), \tilde{y}^{(o)}) \\
&:= \sum_{n=1}^{N}\sum_{o=1}^{O} \mathbb{1}(o \in \mathcal{S}(\mathbf{x}_n)) \cdot \Big[ \text{CE}\big(\hat{\mathbf{A}}_{\phi}^{(o)}(\mathbf{x}_n) \cdot \hat{\mathbf{p}}_{\theta}(\mathbf{x}_n), \tilde{\mathbf{y}}_n^{(o)}\big) + \lambda \cdot \text{tr}\big(\hat{\mathbf{A}}_{\phi}^{(o)}(\mathbf{x}_n)\big) \Big]
\end{aligned}
\tag{4}
$$

where $\mathcal{S}(\mathbf{x})$ denotes the set of all positive coarse labels available for image $\mathbf{x}$, and $\text{tr}(\mathbf{A})$ denotes the trace of the matrix $\mathbf{A}$. The mean trace represents the average probability that a randomly selected annotator provides an accurate label. Intuitively, minimizing the trace encourages the estimated annotators to be maximally unreliable while minimizing the cross entropy ensures fidelity with observed noisy annotators. We minimize this combined loss via stochastic gradient descent to learn both $\{\theta, \phi\}$.

**Learning with Negative Coarse Label.** For some situations, it is easier to provide the negative coarse annotation, e.g., complementary label, to help ML model predict the true label distribution. Thus, we study a readily available substitute, namely complementary labeling. However, if we still use loss functions $\mathcal{L}_{\text{obj}}(\theta, \phi)$ when learning with these complementary labels, similar to Eq. (4), we can only learn a mapping $\mathbb{R} \to \mathcal{Y}$ that tries to predict conditional probabilities $p(\tilde{\mathbf{y}}^{(c)} | \mathbf{x})$ and the corresponding complementary pixels classifier that predicts a $\tilde{y}_{wh}^{(c)}$ for a given observation $\mathbf{x}$.

To address the above issue, inspired by Yu et al. (2018), which summarizes all the probabilities into a transition matrix $\mathbf{M} \in \mathbb{R}^{L \times L}$, where $\mathbf{m}(\mathbf{x},w,h)_{ij} := p(\tilde{y}_{wh}^{\{c\}} = i | y_{wh} = j, \mathbf{x})$ and $\mathbf{m}(\mathbf{x},w,h)_{ii} = 0, \forall i,j \in \{1,...,L\}$. Here, $\mathbf{m}_{ij}$ denotes the entry value in the $i$-th row and $j$-th column of $\mathbf{M}$. As shown in Fig. 1, we achieve this simply by adding a linear layer to the complementary label learning channel. This layer outputs $v(\mathbf{x})$ by multiplying the output of the CE function (i.e., $u(\mathbf{x})$ ) with the transposed transition matrix $\mathbf{M}^{\top}$. Note that the transition matrix is also widely exploited in Markov chains Gagniuc (2017) and has many applications in machine learning, such as learning with label noise Tanno et al. (2019); Zhang et al. (2020b;a).

Recall that in transition matrix $\mathbf{M}$, $\mathbf{m}_{ij} = p(\tilde{y}_{wh}^{\{c\}} = i | y_{wh} = j, \mathbf{x})$ and $\mathbf{m}_{ii} = p(\tilde{y}_{wh}^{\{c\}} = i | y_{wh} = i, \mathbf{x}) = 0$. We observe that $p(\tilde{\mathbf{y}}^{\{c\}} | \mathbf{x})$ can be transferred to $p(\tilde{\mathbf{y}}^{\{c\}} | \mathbf{x})$ by using the transition matrix $\mathbf{M}$,

$$
\begin{aligned}
p(\tilde{y}_{wh}^{(c)} = j | \mathbf{x}) = \sum_{i \neq j} p(\tilde{y}_{wh}^{(c)} = j, \bar{y}_{wh}^{(c)} = i | \mathbf{x}) &= \sum_{i \neq j} p(\tilde{y}_{wh}^{(c)} = j | \bar{y}_{wh}^{(c)} = i, \mathbf{x}) p(\bar{y}_{wh}^{(c)} = i | \mathbf{x}) \\
&= \sum_{i \neq j} p(\tilde{y}_{wh}^{(c)} = j | \bar{y}_{wh}^{(c)} = i) p(\bar{y}_{wh}^{(c)} = i | \mathbf{x})
\end{aligned}
$$

Intuitively, if $\mathbf{v}_i(\mathbf{x})$ tries to predict the probability $p(\tilde{y}^{(c)} = i | \mathbf{x}), \forall i \in [1,...,L]$, then $\mathbf{M}^{-\top}\mathbf{v}$ can predict the probability $p(\tilde{y}^{(o)} | \mathbf{x})$, which is the positive prediction of the corresponding complementary coarse label. To enable end-to-end learning rather than transferring after training, we let

$$
\mathbf{v}(\mathbf{x}) = \mathbf{M}^{\top}\mathbf{u}(\mathbf{x})
$$

where $\mathbf{u}(\mathbf{x})$ is now an intermediate output of the complementary coarse annotation, and $\mathcal{L}_{\text{comp}}(\theta, \phi) = \text{argmax}_{i \in [L]} \mathbf{v}_i(\mathbf{x})$. Then, the modified loss function $\bar{\mathcal{L}}_{\text{obj}}(\theta, \phi)$ is

$$
\begin{aligned}
\bar{\mathcal{L}}_{\text{obj}}(\theta,\phi)(\mathbf{u}(\mathbf{x}), \tilde{y}^{(c)}) &:= \mathcal{L}_{\text{comp}}(\theta,\phi)(\mathbf{v}(\mathbf{x}), \tilde{y}^{(c)}) := \mathcal{L}_{\text{comp}}(\theta,\phi)(\mathbf{M}^{\top} \cdot \{\hat{\mathbf{A}}_{\phi}^{(c)}(\mathbf{x}_n) \cdot \hat{\mathbf{p}}_{\theta}(\mathbf{x}_n)\}, \tilde{y}^{(c)}) \\
&:= \sum_{n=1}^{N}\sum_{c=1}^{C} \mathbb{1}(c \in \mathcal{S}(\mathbf{x}_n)) \cdot \Big[ \text{CE}\big(\mathbf{M}^{\top} \cdot \{\hat{\mathbf{A}}_{\phi}^{(c)}(\mathbf{x}_n) \cdot \hat{\mathbf{p}}_{\theta}(\mathbf{x}_n)\}, \tilde{\mathbf{y}}_n^{(c)}\big) + \lambda \cdot \text{tr}\big(\hat{\mathbf{A}}_{\phi}^{(c)}(\mathbf{x}_n)\big) \Big]
\end{aligned}
\tag{5}
$$

In this way, if we can learn an optimal $\mathbf{v}$ such that $\mathbf{v}_i(\mathbf{x}) = p(\tilde{y}^{(c)} = i | \mathbf{x}), \forall i \in [L]$, meanwhile, we can also find the optimal $\mathbf{u}$ and the loss function $\mathcal{L}_{\text{comp}}(\theta, \phi)$, which can be easily applied to deep learning. With sufficient training examples with complementary coarse labels, this CNN often simultaneously learns good classifiers for both $(\mathbf{x}, \tilde{y}^{(c)})$ and $(\mathbf{x}, \tilde{y}^{(o)})$.

Finally, we combine the positive annotation loss $\mathcal{L}_{\text{obj}}$ and the negative annotation loss $\mathcal{L}_{\text{comp}}$ as our objective and optimize:

$$
\mathcal{L}_{\text{final}}(\theta,\phi) := \mathcal{L}_{\text{obj}}(\theta,\phi) + \mathcal{L}_{\text{comp}}(\theta,\phi).
\tag{6}
$$

## 4 EXPERIMENTS

### 4.1 SET-UP

**MNIST Experiments.** We define a MNIST segmentation dataset to study the properties of the proposed algorithm. While MNIST was originally constructed to facilitate research in recognition (classification) of handwritten digits LeCun et al. (1998), it has found its use in segmentation tasks to segment the digits from the background. It can be seen as an image classification task, except that instead of classifying the whole image, we are classifying each pixel individually. MNIST dataset consists of 60,000 training and 10,000 testing examples, all of which are $28 \times 28$ grayscale images of digits from 0 to 9, and we derive the segmentation labels by thresholding the intensity values at 0.5.

**Cityscapes Experiments.** The Cityscapes Cordts et al. (2016) dataset contains 5000 high-resolution (2048 $\times$ 1024 pixels) urban scene images collected across 27 European Cities. The dataset comprises 5,000 fine annotations (2,975 for training, 500 for validation, and 1,525 for testing) and 20,000 coarse annotations where 11,900 samples for training and 2,000 for validation (i.e., coarse polygons covering individual objects).

**Retinal Image Experiments.** The LES-AV Orlando et al. (2018) is a dataset for retinal vessel segmentation on color fundus images. It comprises 22 fundus photographs with available manual annotations of the retinal vessels including annotations of arteries and veins. The 22 images/patients are acquired with resolutions of 30° field-of-view (FOV) and $1444 \times 1620$ pixels (21 images), and 45° FOV and $1958 \times 2196$ pixels (one image), with each pixel $=6\mu m$. We divide them into 18 images for training and 4 images for testing.

### 4.2 SYNTHETIC NOISY COARSE ANNOTATIONS

We generate synthetic coarse noisy annotations from an assumed expert consensus label on MNIST, Cityscapes and retinal fundus image dataset, to demonstrate the efficacy of the approach in an idealized situation where the expert consensus label is known. We simulate the positive and negative coarse noisy annotations by performing morphological transformations (e.g., thinning, thickening, fractures, etc) on the expert consensus label and background (complementary label), using Morpho-MNIST software Castro et al. (2019). In particular, *positive coarse noisy annotation* is prone to be poor segmentation, which is simulated by combining small fractures and over-segmentation; *negative coarse noisy annotation* always annotates on the background or complementary label using the same approach. To create synthetic coarse noisy labels in the multi-class scenario, we use a similar simulation to create coarse labels on the Cityscapes dataset. We first choose a target class and then apply morphological operations on the provided coarse mask to create the two synthetic coarse labels at different patterns, namely, objective coarse and complementary coarse annotations. We create training data by deriving labels from the simulated annotations. We also experimented with varying the levels of morphological operations on all datasets, e.g., Fig. S2 (in Appendix) shows different ratios with 1, 0.8, 0.5, 0.3 and 0.01 (scribble) compared to the provided coarse labels, to test the robustness of our methods to varying sensitivities to coarse quality.

### 4.3 COMPARISON METHODS AND EVALUATION METRICS

Our experiments are based on the assumption that no expert consensus label is available a priori, hence, we compare our method against multiple weakly-supervised and semi-supervised methods. In particular, we explore our method with ablation studies, e.g., our method without negative coarse annotation; we also consider two state-of-the-art interactive image segmentation algorithms for generating masks from scribbles: GrabCut Rother et al. (2004) and LazySnapping Li et al. (2004), then training FCNs using the masks generated by these methods. Meanwhile, we compare with other weakly-supervised methods using different ways of annotations, e.g., scribble-guided annotations Lin et al. (2016); Zhang & Zhuang (2022), coarse annotations Saha et al. (2022); Wang et al. (2021), and box-level annotations Dai et al. (2015); Jiang et al. (2022). Finally, we compare our weakly-supervised results based on the noisy coarse annotations and strongly-supervised results based on the expert consensus annotations. For evaluation metrics, we use mean Intersection over Union (mIoU) between estimated segmentation $\hat{\mathbf{p}}_\theta(\mathbf{x})$ and expert consensus label $\mathbf{y}_{GT}$.

### 4.4 SEGMENTATION PERFORMANCE

**Strategies of utilizing coarse annotation.** Our method jointly propagates information into unmarked pixels and learns network parameters. An easy way is to first use any existing interactive image

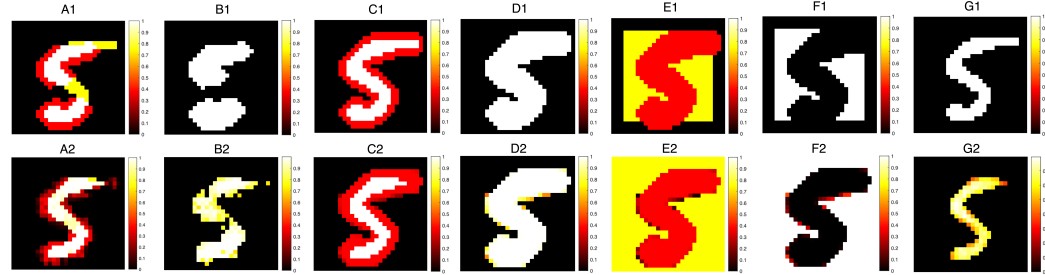

Figure 2: Visualisation of the estimated labels, the estimated pixel-wise CMs, and the estimated TMs on MNIST datasets along with their targets (best viewed in color). White is the true positive, yellow is the false negative, red is the false positive, and black is the true negative. A1-A2: the target and estimated CM: $\mathbf{A}_\phi^{(o)}(\mathbf{x})$ for positive coarse annotation; B1-B2: the given and estimated positive coarse annotation $\tilde{\mathbf{y}}^{(o)}$; C1-C2: the target and estimated intermediate CM: $\mathbf{A}_\phi^{(c)}(\mathbf{x})$ for negative coarse annotation; D1-D2: the target and estimated intermediate negative coarse annotation $\mathbf{u}(\mathbf{x})$; E1-E2: the target and estimated TM: $\mathbf{M}^\top$ for negative coarse annotation; F1-F2: the provided and estimated negative coarse annotation ($\tilde{\mathbf{y}}^{(c)}$ and $\mathbf{v}(\mathbf{x})$); G1-G2: target label and our estimation.

| Methods | MNIST | Cityscapes |
|---|---|---|
| GrabCut + FCN | $75.2 \pm 0.3$ | $53.6 \pm 0.4$ |
| LazySnapping+FCN | $78.5 \pm 0.2$ | $59.4 \pm 0.4$ |
| Ours (w/o negative annotation) | $77.2 \pm 0.2$ | $62.3 \pm 0.2$ |
| Ours (positive and negative annotations) | $82.5 \pm 0.1$ | $68.3 \pm 0.2$ |

Table 1: Segmentation results (mIoU (%)) on the MNIST and Cityscapes validation set via different strategies of utilizing coarse annotations.

| Methods | Annotations | MNIST | Cityscapes |
|---|---|---|---|
| ScribbleSup Lin et al. (2016) | Scribble | $73.4 \pm 0.2$ | $61.3 \pm 0.3$ |
| CycleMix Zhang & Zhuang (2022) | Scribble | $75.2 \pm 0.4$ | $63.2 \pm 0.3$ |
| CoarseSup Saha et al. (2022) | Coarse | $76.6 \pm 0.4$ | $62.6 \pm 0.4$ |
| LC-MIL Wang et al. (2021) | Coarse | $76.8 \pm 0.2$ | $64.3 \pm 0.2$ |
| BoxSup Dai et al. (2015) | box | $78.5 \pm 0.3$ | $64.8 \pm 0.5$ |
| L2G Jiang et al. (2022) | box | $79.2 \pm 0.2$ | $65.4 \pm 0.4$ |
| Ours | scribble | $75.3 \pm 0.3$ | $62.7 \pm 0.5$ |
| Ours | coarse | $82.5 \pm 0.2$ | $68.3 \pm 0.2$ |

Table 2: Comparisons (mIoU (%)) of weakly-supervised methods on the MNIST and Cityscapes validation set, using different ways of annotations.

segmentation methods to generate masks based on coarse annotation, and then use these masks to train FCNs. In Table 1, we compare our methods with these two-step solutions.

We investigate two popular interactive image segmentation algorithms for generating masks from coarse annotation: GrabCut Rother et al. (2004) and LazySnapping Li et al. (2004). Given the coarse annotations, GrabCut generates the mask only for the target pixels while LazySnapping produces the masks not only for the objective but also for the non-target pixels. Training FCNs using the masks generated by these methods shows inferior semantic segmentation accuracy. This is because these traditional methods Rother et al. (2004); Li et al. (2004) only focus on the low-level color/spatial information and are unaware of semantic content. The generated masks cannot be the reliable "GT" for training the supervised networks.

On the contrary, our coarse-based supervised method achieves a score of 82.5% on MNIST and 68.3% on Cityscapes dataset, about 10% higher than the two-step solutions. This is because our model can capture the patterns of mistakes for each noisy coarse annotation, and the high-level information can help with the coarse-to-fine propagation of the true label estimation. This behavior is shown in Fig. 2.

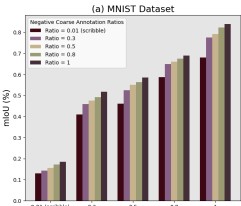 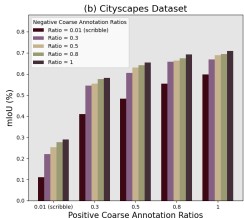 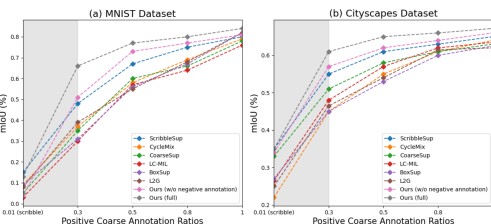

Figure 3: Sensitivities to quality of positive and negative coarse annotations. The smaller coarse areas are synthesized from the provided coarse annotations, reducing their areas by a ratio.

Figure 4: Comparison of sensitivities to quality of positive coarse annotations evaluated on different weakly-supervised methods. The smaller coarse areas are synthesized from the provided coarse annotations, reducing their areas by a ratio.

| supervision | w/ masks | w/ coarse | total | mIoU (%) | supervision | w/ masks | w/ coarse | total | mIoU (%) |
|---|---|---|---|---|---|---|---|---|---|
| weakly | – | 400 | 400 | $82.5 \pm 0.2$ | weakly | – | 174 | 174 | $68.3 \pm 0.2$ |
| strongly | 400 | – | 400 | $86.2 \pm 0.1$ | strongly | 174 | – | 174 | $71.7 \pm 0.1$ |
| semi | 400 | 200 | 600 | $88.7 \pm 0.2$ | semi | 174 | 87 | 261 | $73.3 \pm 0.1$ |

Table 3: Comparisons of our method using different annotations on the MNIST (left) and Cityscapes (right) validation sets. The term "w/ masks" shows the number of training images with mask-level annotations, and "w/ coarse" shows the number of training images with coarse annotations. The visualization of segmentation results are shown in Figs. S3 and S4 in Appendix.

**Comparisons with other weakly-supervised methods.** In Table. 2, we compared with other weakly-supervised methods using different ways of annotations. We note that while scribble annotations are the most economical, their weakly-supervised learning accuracy (e.g., 75.2% of CycleMix Zhang & Zhuang (2022)) lags behind other ways of annotations. Even though our method is trained with scribble annotation, the performance is still worse than other models. This is because the scribble annotations didn't provide enough information to learn the CMs for correcting the label noise. On the other hand, provided with coarse annotations, our model can achieve significant improvement on par with box-supervised methods (64.8% of BoxSup Dai et al. (2015) and 65.4% of L2G Jiang et al. (2022) on Cityscapes dataset), indicating that coarse annotations can be well exploited with CMs to boxes.

**Comparisons with using masks.** In Table. 3, we compare our weakly-supervised results based on the noisy coarse annotations and strongly-supervised results based on pixel-level annotations. When replacing all mask-level annotations with noisy coarse annotations, our method has a degradation of about 5 points on both MNIST and Cityscapes datasets. We believe this is a reasonable gap considering the challenges of exploiting coarse annotations and the noise inside.

Our method can also be easily generalized to SSL which uses both mask annotations and coarse annotations. For the training of models with weak and strong supervision, we applied an equal number of training samples with noisy coarse annotations and mask-level annotations. To evaluate our SSL results, we use the same number of mask-level annotations and half number of noisy coarse annotations. Our SSL results are 88.7% on MNIST and 73.3% on Cityscapes dataset, showing a gain of approximate 2% higher than the strongly-supervised approach. This gain is due to the extra coarse annotations. Overall, our SSL results are on par with their strongly-supervised result, but we only use a small number of coarse-level annotations as the extra data.

## 4.5 SENSITIVITIES TO THE QUALITY OF COARSE ANNOTATIONS

Due to the different experiences and behaviors of the annotators, the coarse annotations show variabilities in quality. So we investigate how sensitive our method is to the coarse label quality. Generating different sets of coarse annotations usually requires extra annotations efforts, therefore we utilize the provided mask-level labels to create coarse annotations with a variety of qualities. We focus on the area of the coarse annotations. Given any mask-level annotation, we reduce the area of this coarse label by a ratio.

| (a) Sensitivities to annotation quality | (b) Input image | (c) ground-truth | (d) semi-supervised | (e) strongly-supervised | (f) coarse-supervised |

Figure 5: Sensitivities to quality of positive and negative coarse annotations, and the visualization of the estimated labels with different supervision approaches.

In terms of choosing the different ratios for the coarse labels, the central point is randomly chosen from the original mask-level annotation, and the area is determined by the reduced ratio. Fig. S2 in Appendix shows examples of the reduced coarse annotations, where a ratio of 0.01 means a scribble is used.

Fig. 3 shows the results of our coarse-supervised algorithm using coarse annotations of different ratios. Our method generates much better results when the coarse label area increased from 0 to 0.3, and the results from Fig. 4 show that our method performs gradually better when the coarse annotation areas are increased, suggesting that our method has reasonably learned the useful information and corrected the error from the coarse annotation with different qualities.

### 4.6 EXPERIMENTS ON RETINAL VESSEL SEGMENTATION

Lastly, we illustrate the results of our approach on a more challenging dataset with real coarse and noisy labels from the medical domain. This dataset, called LES-AV, consists of images of the retinal fundus acquired from different patients. The task is to segment the vessel into a binary mask (see Fig. 5). The process of segmenting the blood vessel in the retinal image is crucial for the early detection of eye diseases.

An experienced annotator was tasked with providing the practical positive and negative coarse annotations for each sample on LES-AV dataset. We generate such a real-world dataset to show the segmentation results and evaluate the performance of different supervision approaches. Meanwhile, we also created 5 different ratio levels for the positive and negative coarse annotations from *level-1* (tend to scribble) to *level-5* (tend to GT) with increasing ratios compared to the given expert consensus labels. We use such a dataset to evaluate the sensitivity to annotation quality of our model on medical image data.

We show the results of sensitivities to annotation quality in Fig. 5(a). Similarly to the previous experiments, our model performs robustly and gradually improved when the ratio of coarse annotations is increased. Especially when the ratio is increased from *level-1* to *level-2*, our model's performance is increased significantly and comparable to the mask-level results. By applying our practical annotations, we conduct a group of experiments under different supervision. The results in Table. S2 indicate that our WSL approach achieves comparable results to the strongly-supervised method. Meanwhile, by including some extra coarse annotations, the result is improved 3%. Finally, we present the segmentation visualization in Fig. 5(d∼f).

## 5 DISCUSSION AND CONCLUSION

We introduced a new theoretically grounded algorithm for recovering the expert consensus label distribution from noisy coarse annotations. Our method brings implementation simplicity, requiring only adding a complementary label learning term to the loss function. Experiments on both synthetic and real data sets have shown superior performance over the common WSL and SSL methods in terms of both segmentation accuracy and robustness to the quality of coarse annotations and label noise. Furthermore, the method is capable of estimating coarse annotations even when scribble is given per image.

Our work was primarily motivated by medical imaging applications for which pixel-level annotation is required for a large number of images. However, future work shall consider imposing structures on the CMs and TMs to broaden the applicability to scribble or spot annotations, which could contribute to saving the labor of the labeling process. Another limiting assumption is to learn only from the coarse annotation in difficult cases. The majority of segmentation failures happen in the difficult cases or the difficult patches in the image. Only giving coarse annotations on these difficult cases or patches and learning from them is also a valuable next step.

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
