## APPENDIX

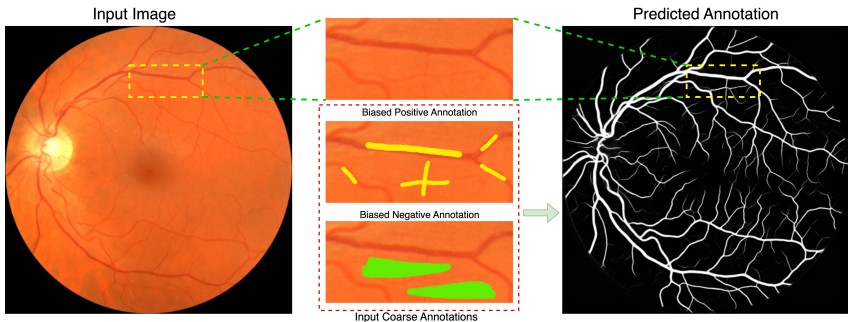

Figure S1: Illustration of a segmentation example using the noisy positive and negative coarse annotations.

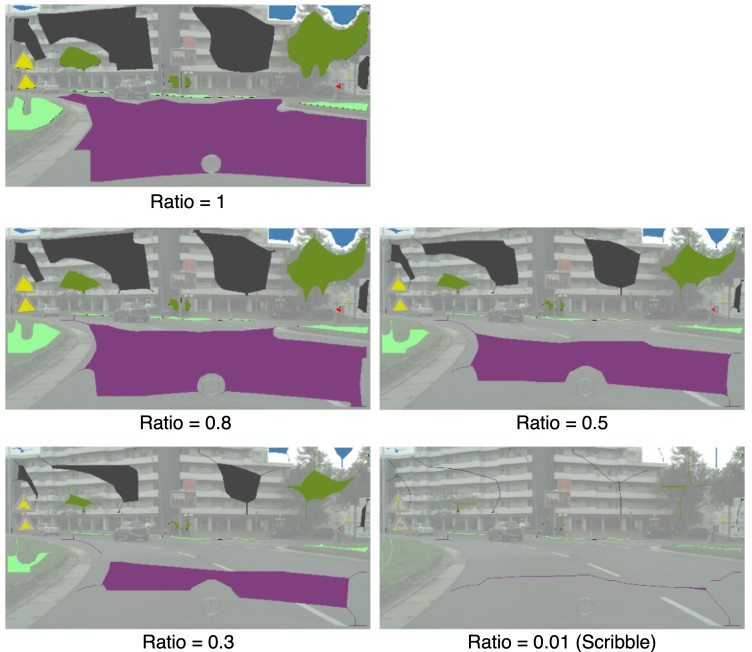

Figure S2: Coarse annotations of different ratios for investigating our method's sensitivities to coarse quality.

## A    IMPLEMENTATION DETAILS

Our method is implemented in Pytorch Fey & Lenssen (2019). Our network is based on a 4 down-sampling stages 2D U-net Ronneberger et al. (2015), the channel numbers for each encoders are 32, 64, 128, 256, we also replaced the batch normalisation layers with instance normalisation. Our segmentation network and coarse annotation network share the same parameters apart from the last layer in the decoder of U-net, essentially, the overall architecture is implemented as an U-net with multiple output last layers: one for prediction of true segmentation; others for predictions of positive/negative coarse segmentation respectively. For segmentation network, the output of the last layer has c channels where c is the number of classes. On the other hand, for coarse annotation network, by default, the output of the last layer has $L \times L$ number of channels for estimating confusion matrices at each spatial location. All of the models were trained on a NVIDIA RTX 208 for at least 3 times with different random initialisations to compute the mean performance and its standard deviation (run 3 times of the experiments with the same initialization). The Adam Kingma & Ba (2014) optimiser was used in all experiments with the default hyper-parameter settings. We also provide all of the hyper-parameters of the experiments for each data set in Table S1. We also kept the training details the same between the baselines and our method.

| Data set | Learning Rate | Epoch | Batch Size | Augmentation | weight for regularisation ($\lambda$) |
|---|---|---|---|---|---|
| MNIST | 1e-4 | 131 | 4 | Random flip | 0.8 |
| Cityscapes | 1e-4 | 72 | 16 | Random flip | 1.5 |
| LES-AV | 1e-4 | 55 | 16 | Random flip | 1.5 |

Table S1: Hyper-parameters used for respective datasets.

## B ADDITIONAL RESULTS

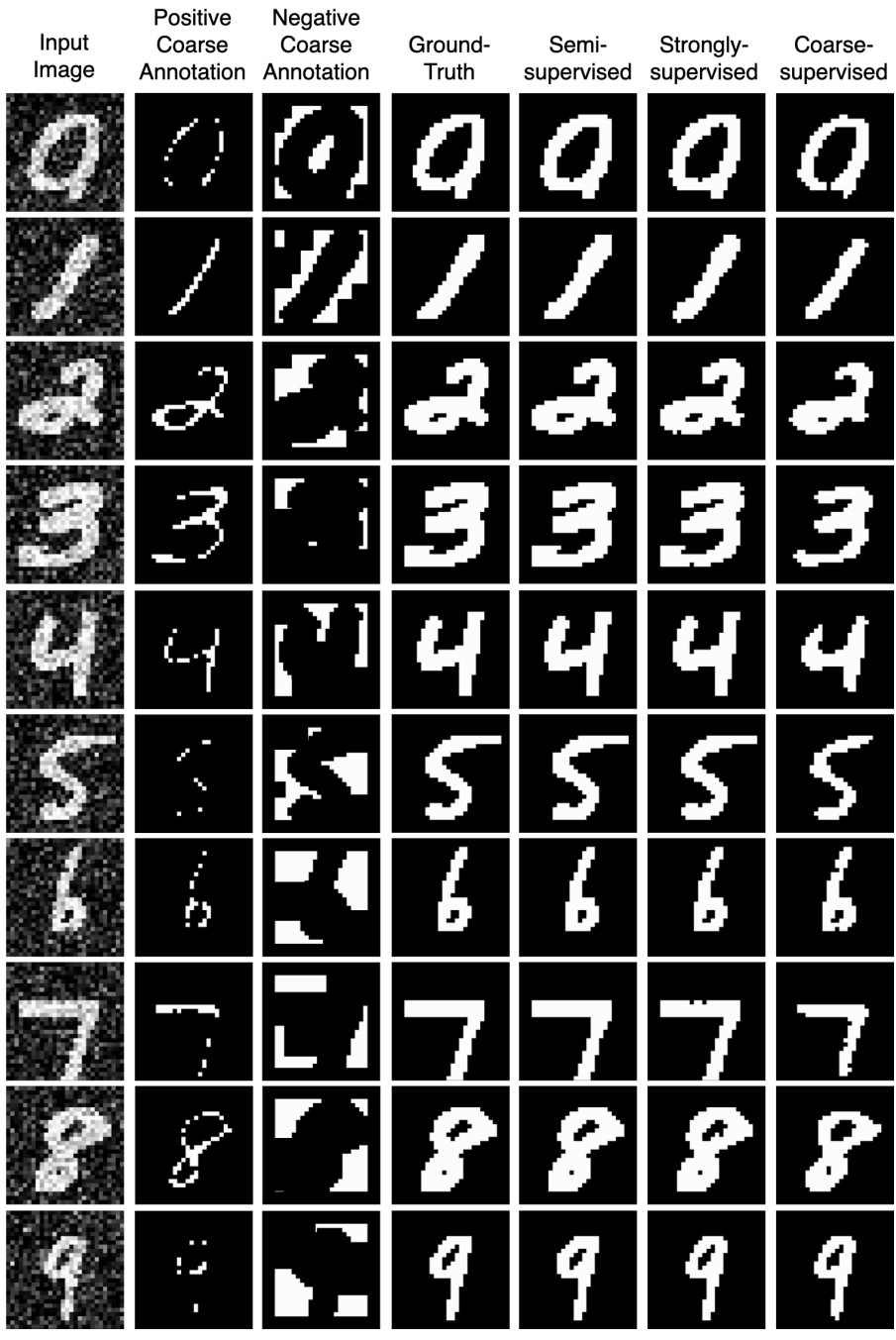

Figure S3: Visualization of the segmentation results on MNIST dataset.

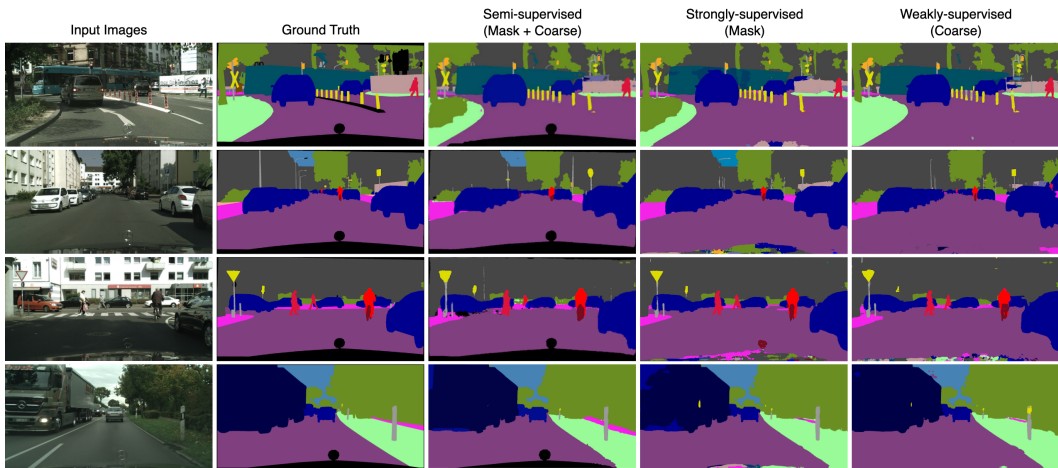

Figure S4: Visualization of the segmentation results on Cityscapes dataset.

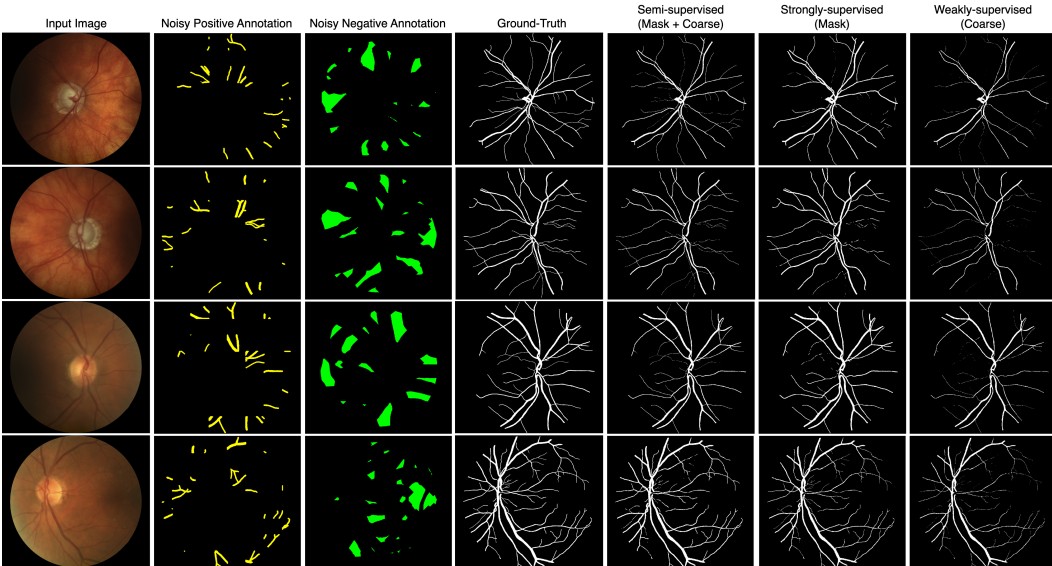

Figure S5: Visualization of the vessel segmentation results on the retinal image dataset.

| supervision | w/ masks | w/ coarse | total | mIoU (%) |
|---|---|---|---|---|
| weakly | — | 16 | 16 | $65.8 \pm 0.3$ |
| strongly | 16 | — | 16 | $69.2 \pm 0.2$ |
| semi | 16 | 4 | 20 | $71.6 \pm 0.3$ |

Table S2: Comparisons of our method using different annotations on the LES-AV retinal image validation set. The term "w/ masks" shows the number of training images with mask-level annotations, and "w/ coarse" shows the number of training images with coarse annotations. More visualization of segmentation results is shown in Fig. S5.