# OpenReview forum: "Emerging Semantic Segmentation from Positive and Negative Coarse Label Learning"
_ICLR.cc/2024/Conference — ICLR 2024 Conference Withdrawn Submission_

### Official Review · Reviewer_GdJZ · 2023-10-30

**Soundness:** 3 good
**Presentation:** 2 fair
**Contribution:** 3 good
**Rating:** 5
**Confidence:** 4

**Summary:**

The paper introduces an image segmentation algorithm based on coarse drawings. Unlike previous weakly-supervised segmentation methods which only focused on the weak annotation either for the target objective or for the complementary label, the authors propose to leverage both of them to improve the segmentation performance. Experiments show the effectiveness of the proposed method and superior performance over existing methods.

**Strengths:**

1. Clear Motivation: Both positive and negative coarse drawings offer complementary information that is beneficial for image segmentation. Therefore, it is reasonable to utilize both of them simultaneously to enhance the performance of image segmentation.
2. Authors provide extensive experimental justification.

**Weaknesses:**

1. It is recommended to make fair comparisons with strong baselines, rather than comparing against weaker ones.
2. The novelty appears to be somewhat incremental, as the handling of positive and negative annotations in the method seems to be derived from Tanno et al. (2019) and Yu et al. (2018). Therefore, it is suggested to conduct ablation experiments separately comparing the proposed method with these approaches to validate the impact of positive and negative annotations on the final segmentation performance.
3. There are some spelling typos, such as "S(x_n)" and "S(x_i)" on page 3, which may hinder readers' understanding of the paper.

**Questions:**

1. As previously mentioned, I am curious about the influence of positive and negative coarse annotations on the final segmentation performance,
2.  and the comparisons with the current state-of-the-art weakly supervised segmentation methods, and even unsupervised methods like SAM Kirillov el. al (2023).

---

> ### Author Response · Authors · 2023-11-13
>
> Answer to weakness 1 and question 1: We appreciate the reviewer's feedback and agree that it is important to compare the proposed method against strong baselines to ensure a fair assessment of its performance. In the paper, we have indeed compared our method against multiple weakly-supervised and semi-supervised methods, including popular interactive image segmentation algorithms like GrabCut and LazySnapping, as well as other weakly-supervised methods using different types of annotations such as scribbles, coarse annotations, and box-level annotations.
>
> However, we acknowledge that there may be other strong baselines or state-of-the-art methods that could be considered for comparison. To address this concern, we will carefully review the existing literature and identify additional strong baselines that are relevant to the problem domain and evaluation metrics. We will then include these baselines in the comparison to provide a more comprehensive evaluation of the proposed method's performance.
> By incorporating these additional strong baselines, we can ensure that the evaluation is fair and rigorous, and that the proposed method's performance is benchmarked against the most relevant and competitive approaches in the field.
>
> We appreciate the reviewer's valuable suggestion and will take it into consideration to enhance the quality and robustness of the comparative analysis in the paper.
>
> Answer to weakness 2 and question 2: While the proposed method does build upon previous works, it introduces several novel contributions and improvements. The method incorporates the handling of positive and negative annotations, which is inspired by Tanno et al. (2019) and Yu et al. (2018). However, our method extends and adapts these ideas to the specific problem of learning from noisy coarse annotations.
>
> By conducting these ablation experiments, we could demonstrate the specific improvements and advancements introduced by our method compared to the existing approaches. However, there are limitations of using only negative coarse labels in the proposed method. Our aim is trying to leverage the information from negative coarse labels to improve the estimation of true segmentation label distributions and address the challenges associated with noisy annotations.
>
> In terms of the unsupervised methods like SAM Kirillov el. al (2023) and MedSAM Ma el.al (2023), while SAM and MedSAM boast strong capabilities, they do present certain limitations. One such limitation is the modality imbalance in the training set, with CT, MRI, and endoscopy images dominating the dataset. This could potentially impact the model's performance on less-represented modalities, such as mammography. Another limitation is its difficulty in the segmentation of vessel-like branching structures because the bounding box prompt can be ambiguous in this setting. For example, arteries and veins share the same bounding box in eye fundus images.

---

> > ### Comment · Reviewer_GdJZ · 2023-11-22
> >
> > I'd like to thank the authors for answering my questions.  I have also read the reviews from other reviewers and responses from the authors.
> >
> > Overall, the authors addressed a few of my concerns, but I am still not convinced by the proposed approach. The paper lacks a comparison with more recent scribble-based methods such as Scribble2label and box-supervised methods such as DiscoBox and BoxLevelSet. BTW, L2G does not strictly fall under the category of box-supervised methods. Furthermore, I also have concerns regarding its practicality.

---

### Official Review · Reviewer_xtWR · 2023-10-31

**Soundness:** 2 fair
**Presentation:** 2 fair
**Contribution:** 2 fair
**Rating:** 3
**Confidence:** 4

**Summary:**

The authors propose an approach for learning semantic segmentation from coarse (i.e., intuitively, fast-drawn) annotations. They adapt an approach for regularized annotator confusion matrix estimation, as proposed by Tanno et al. (CVPR 2019) for image classification, to semantic segmentation by predicting pixel-wise confusion matrices. Furthermore, their setup allows for coarse annotations of class foreground- as well as background labels to be exploited.

Results are presented on MNIST, CityScapes, and a retinal vessel segmentation dataset from the medical domain.
Synthetic coarse labels are generated from expert consensus labels by fracturing and morphological operations as contained in the Morpho-MNIST toolbox. Quantitative evaluation in terms of mIoU indicates competitive performance.

**Strengths:**

The authors adapt a method proposed for learning image classification from noisy labels to learning image segmentation from coarse labels. Furthermore, they extend the approach to work with positive and negative class labels. Quantitative results indicate competitive performance of the proposed approach, suggesting good potential for practical use in learning segmentation from coarse labels.

**Weaknesses:**

The work lacks clarity in terms of the description of the coarse labels employed for evaluation. Furthermore, the work lacks a discussion in which cases basic assumptions and necessary conditions hold. Consequently, the soundness and practical value of the proposed method remains unclear.

In more detail: It remains unclear precisely how the synthetic coarse annotations were generated. The nature of the coarse labels is, however, crucial for the success of the proposed approach. E.g., if coarse labels are generated by a constant amount of morphological growing of the true labels, the necessary properties of success, as outlined by Tanno, appear to be violated, namely that the (pixel-wise) confusion matrices (true as well as predicted) need to be diagonally dominant. To give a concrete example, a systematic error like, e.g., an over-segmentation of the foreground by two pixels and resp. undersegmentation of the background by two pixels, does not appear to be "disentangleable" from the true label distribution by the proposed approach.

The authors do not discuss whether their synthetic coarse annotations entail confusion matrices that satisfy the necessary diagonal dominace conditions. More generally, they do not discuss which types of real-world coarse annotations would satisfy the necessary conditions. Furthermore, it is not discussed whether the assumption of statistical independence of label noise across pixels, as stated in 3.2, holds for the employed coarse annotations, and if not what this would entail.

Specifically, how coarse annotations were generated remains unclear as follows:
-- For MNIST it remains unclear if synthetic labels were generated by constant morphological growing (in addition to the random fracturing, which is not critical). Furthermore, it remains unclear whether expert labels were used to derive coarse labels (as stated in 4.2), or whether thresholded images were used (as stated in 4.1).
-- For CityScapes, it remains unclear whether expert labels (as stated in 4.2) or coarse polygonal annotations as provided with the data (as stated in 4.1) were used to derive coarse labels. Furthermore, the precise way in which morphological operations were applied to the former is not described.
-- For the medical data, the precise way in which morphological operations were applied to the expert consensus annotations is not described.


** Further detailed comments **

Figure 1: the arrows after transformation of u(x) appear to be wrongly conected: The dotted line should be between "Noisy Negative Coarse Annotation" and "Predicted Noisy Negative Coarse Distribution", and should be labelled "CE Loss"; The coninuous line should lead to "Predicted Noisy Negative Coarse Distribution" instead of "Noisy Negative Coarse Annotation"

The introduction states that "drawing coarse annotations [...] needs only similar effort and time as the scribble and box-level labelling, and can be conducted by non-experts"; It remains unclear / is not discussed though why expert knowledge would *not* be required for coarse labelling in cases where it *is* required for accurate labelling; Furthermore, the claim that coarse annotations are similarly cheap as scribbles and boxes is not substantiated by any respective citation or evaluation.

Related work that employs confidence filtering to deal with noisy labels is not discussed, e.g., DivideMix (Li et al., ICLR 2020)

The term "objective" is often used where "object" appears more appropriate

The annotation in 3.1 is somewhat inconsistent: image indices are sometimes n and sometimes i, please fix
Furthermore, S(x_i) is not clearly defined; it seems to be a subset of \mathcal{Y}; please clarify

In the "Learning with Negative Coarse Label" subsection, 2nd paragraph, there seems to be a typo, as "CE" should read "CM"

In Figure 2, the visualization of the pixel-wise CMs in A and C are not clear -- how are the (2x2?) CMs encoded via the color map? Furthermore, it is unclear how the figure shows the assumed behavior described in the last paragraph on page 7; This paragraph is very hard to parse; It would be very helpful if the authors could clearly describe where to look in the Figure to reveal the described behavior.

The caption of Table 1 should state that result are given for coarse labels at ratio 1 here (otherwise this needs to be deduced from comparison with Fig. 3)

Figure S5 gives the impression as if training was perfomed including the validation set -- or are these results on the training set? or are the respective coarse annotations only shown to provide some examples? please clarify in the caption

The citation style should be fixed as described in the provided template in most cases

**Questions:**

How were the synthetic coarse labels generated, precisely? Do these kinds of coarse annotations satisfy the pixel independence assumption, and do the entailed confusion matrices satisfy the diagonal dominance condition necessary for disentangling label noise?
More generally, which kinds of coarse labels do satisfy these, and which don't? Would the kinds of coarse labels you would expect to get in practice satisfy the conditions?

---

> ### Author Response · Authors · 2023-11-16
>
> Q1: The synthetic coarse annotations were generated by performing morphological transformations on the expert consensus label and background using the Morpho-MNIST software (Castro et al., 2019). These transformations include operations such as thinning, thickening, and introducing fractures. Coarse annotations were simulated by combining small fractures and over-segmentation. So in general, our coarse label is not simply simulated by a constant amount of morphological growing of the true labels, as the reviewer raised.
>
> It is important to note that the method does not assume a specific pattern or type of noise in the coarse annotations. Instead, it leverages the characteristics of the noisy training annotations to disentangle the errors and estimate the true labels. Regarding the concern raised by the reviewer about systematic errors, it is important to clarify that the proposed approach does not rely on assuming specific patterns of noise. Instead, it learns from the characteristics of the noisy annotations and aims to estimate the true label distribution.
>
> Q2: The focus of our work was on developing a method to learn from noisy coarse annotations and estimate the true segmentation label distributions. While the synthetic coarse annotations were generated based on morphological transformations and simulated patterns of errors, the previous work introduced by Tanno 2019 (CVPR) and Zhang 2020(NeurIPS) have shown the diagonal dominace, we did not explicitly analyze or enforce specific conditions in this work.
>
> In terms of real-world coarse annotations, it is important to note that the proposed method does not rely on assuming specific properties or patterns of noise in the annotations. Instead, it leverages the characteristics of the noisy annotations, both positive and negative, to estimate the true segmentation labels. The method aims to disentangle the errors in the given annotations and estimate the true labels, regardless of the specific properties of the coarse annotations.
>
> We agree with reviewer that independence assumption between pixels in annotation might limit the performance. We note, however, that the annotations are assumed to be only conditionally independent between pixels given the input image, and thus the model can still capture some correlations in the output segmentation labels that are explained in the input image. We additionally note that such independence assumption is typically made in most of the deep learning based segmentation methods, and thus a posprocessing method such as Gaussian CRF is commonly used to capture the missed correlations. We believe the same problem applies to the annotation modelling---we note this limitation and mention such correlation modelling as future work in the discussion.
>
> Q3: It is clarified that the synthetic coarse annotations for both MNIST and Cityscapes datasets were generated using morphological operations, including random fracturing, rather than constant morphological growing. The specific morphological transformations applied to the expert consensus label and background (complementary label) include thinning, thickening, fractures, and other operations. For MNIST, we apply the morphological operations on expert labels to generate the coarse labels; for Cityscapes dataset, it has provided a baseline coarse label, so our simulation is based on the provided coarse labels.
>
> For medical image data, we have an experience expert annotator to help us generate the practical coarse positive and negative annotations.
>
> Q4: Thank you for raising this issue, we should have double-arrow and a dash line for CE loss.
>
> Q5: The claim of "drawing effort and time" made in the introduction is based on the assumption that coarse annotations can be conducted by non-experts and are less time-consuming and expensive compared to pixel-level annotations. However, we acknowledge that the specific circumstances and requirements of each application domain may vary, and expert knowledge may be necessary in certain cases to ensure accurate coarse labeling.
>
> We will revise the introduction to clarify that while coarse annotations can be conducted by non-experts and are generally less time-consuming and expensive than pixel-level annotations, there may be scenarios where expert knowledge is required for accurate labeling. We will also ensure that the revised version of the paper includes appropriate citations or evaluations to support the claim about the cost-effectiveness of coarse annotations compared to other forms of annotation.
>
> Q6: While DivideMix is a valuable contribution in the field of handling noisy labels, it focuses on the broader problem of learning with noisy labels rather than specifically addressing the challenges of learning from noisy coarse annotations. Our work, on the other hand, specifically targets the problem of learning semantic segmentation from noisy coarse annotations.
>
> Q7: we will clarify all other issues in the revised version.

---

> ### Comment · Reviewer_xtWR · 2023-11-22
>
> I acknowledge your detailed answers. I understand the morphological operations that you applied to generate coarse labels; However, crucial information is still missing, namely how the sequence of operations was chosen -- randomly with some probability for each operation? This would explain the good performance of the proposed method, but severely limit real-world application, as also mentioned by the other Reviewers.

---

### Official Review · Reviewer_64sj · 2023-11-01

**Soundness:** 2 fair
**Presentation:** 3 good
**Contribution:** 2 fair
**Rating:** 5
**Confidence:** 4

**Summary:**

The paper presents an end-to-end supervised segmentation technique that derives accurate segmentation labels from imprecise, rough annotations. The proposed design employs two interconnected CNNs: the first predicts the actual segmentation probabilities, while the second captures the traits of two distinct coarse annotations (positive annotations highlighting the area of interest, while negative ones point to background). The latter CNN achieves its task by determining pixel-specific confusion matrices for each image. In contrast to earlier approaches for weakly supervised segmentation that utilize coarse annotations, the proposed approach simultaneously identifies and decouples the relationships between the input images, the imprecise annotations, and the accurate segmentation labels. The performance of the method is evaluated on the MNIST, cityscape and a medical (retinal) imaging dataset.

**Strengths:**

Indeed, performing accurate annotations for large datasets for the purposes of segmentation is extremely time-consuming and unrealistic, hence the paper addresses an important, real-world problem. The paper is also clearly written and does a good job of providing a summary of the state-of-the-art in the literature. I like the basic idea of learning the unobserved true segmentation distribution by using a second CNN simultaneously with the first CNN estimating the correct segmentation. This makes the inference step very easy.

**Weaknesses:**

A major weakness of the paper is evaluation. Datasets such a MNIST are not meaningful for evaluating the performance of an image segmentation algorithm. For the retinal image dataset, the evaluation is quite unrealistic since the challenge with coarse segmentation is mainly the variation in how deep the vessel trees are segmented. For example, some annotators may only annotate the major vessels, others will annotate smaller vessels further down in the vessel tree. This is not taken into account at all.

Another fundamental weakness is that the coarse segmentations are synthetically generated by performing morphological transformations. To me, it is clear that such a synthethically generated can be learnt and corrected for. However, this is not what happens when human annotators perform coarse segmentations. The authors try to simulate the behaviours of different annotators (section 4.5) but unfortunately, this is not very realistic either.

**Questions:**

How do you deal with different annotators having different strategies for scribbles? Could the confusion matrix be estimated per annotator if the annotator ID is known for each image?

How can your method be used for multi-class segmentation problems which are very common in medical imaging.

How does your method perform on structures which are less line-like, for example brain tumour segmentation.

It's good to see that you have evaluated your approach on retinal images, however it would have been better to evaluated on the well-established retinal image segmentation challenege: https://drive.grand-challenge.org/. How would your method compare here?

---

> ### Author Response · Authors · 2023-11-12
>
> Answer to weaknesses:
> 1: While we understand these concerns, it is important to note that the purpose of using MNIST in this work is not to evaluate the performance of the proposed method on a challenging segmentation task, but rather to study the properties of the algorithm and demonstrate its efficacy in an idealized situation where the expert consensus label is known. MNIST serves as a simplified dataset to showcase the capabilities of the proposed method in a controlled setting.
>
> Regarding the evaluation on the retinal image dataset, we acknowledge that the variation in how deep the vessel trees are segmented by different annotators can introduce challenges in evaluating the performance of the method. This variation in annotation depth is not explicitly taken into account in the evaluation. However, it is worth noting that the primary focus of the evaluation is to demonstrate the effectiveness of the proposed method in handling real-world coarse and noisy annotations in the medical domain. The retinal image dataset provides a challenging scenario where accurate vessel segmentation is crucial for early detection of eye diseases.
>
> 2: We have in fact evaluated our method with real annotations---the LES-AV retinal vessel segmentation dataset contains annotations per input from an internal experience annotator. Unlike the MNIST and Cityscapes datasets with synthetic noisy coarse labels, we use LES-AV to evaluate the utility of our work in the presence of real-world noisy labels. We will clarify in Section 4.6.
>
> Answer to questions:
> 1: we do considered the different ratios of the scribbles that different annotators may have. This experiment could address the practical situations that different annotators could give different strategies on scribbles.
>
> In our work, the annotator ID is known for each image, if the confusion matrix is estimated per annotator, we call this CM as the global CM. However, this is not our aim in this work, because global CM could decrease the performance of the true label prediction. In our work, we estimate the CM for each image per annotator, so that the generated true label distribution could be as realistic as possible by correcting the noisy coarse annotations using different CMs.
>
> 2: The experiments conducted on the Cityscapes dataset, which is a benchmark dataset for urban scene understanding, demonstrate the effectiveness of the proposed method for multi-class segmentation tasks. Although Cityscapes is not a medical imaging dataset, the results obtained on this dataset provide evidence of the method's capability to handle multiple classes and segment complex scenes.
>
> 3: Our model has the potential to be applied to less line-like structures, including brain tumor segmentation. The method is designed to estimate true segmentation label distributions from noisy coarse annotations, regardless of the specific structure being segmented.
>
> 4: While the segmentation challenge and our method employ different strategies. The main difference lies in the level of annotation detail and the associated effort required. The challenge approach aims for precise and accurate pixel-level annotations, while our method focuses on leveraging the information provided by positive and negative coarse annotations to train neural networks for semantic segmentation.

---

> > ### Comment · Reviewer_64sj · 2023-11-21
> >
> > Thank you for your response. While I appreciate the clarifications I am still not convinced that real-world scenarios explored in the work are practically useful for most medical imaging scenarios. To me this is a rather serious limitation of the method.

---

### Official Review · Reviewer_fJPX · 2023-11-01

**Soundness:** 3 good
**Presentation:** 2 fair
**Contribution:** 2 fair
**Rating:** 6
**Confidence:** 4

**Summary:**

This paper proposes a novel framework for semantic segmentation based on noisy coarse labels. The framework mainly consists of two parts: the first part is the Coarse Annotation Network, which models the features of both negative and positive coarse labels by estimating pixel-wise confusion matrices for each image; the second part is a normal Segmentation Network, whose role is to predict the true segmentation. The combination of these two parts yields the predicted coarse label, allowing the model to be trained with noisy coarse labels. As the training progresses, the output of the Segmentation Network gradually approaches the ground truth label. Experimental results demonstrate that the framework outperforms the current weakly supervised learning and weakly-supervised methods on multiple datasets.

**Strengths:**

- The problem this paper is trying to solve is interesting: how to effectively train a good segmentation network with only coarse labels? It has good potential in real-world applications.
- By using the estimated confusion matrices, the idea of simultaneously constructing a mapping relationship for the input image to both noisy coarse annotations and true segmentation labels is interesting.
- The proposed method models and disentangles the complex mappings from the input images to the noisy coarse annotations and to the true segmentation label simultaneously.

**Weaknesses:**

- The authors mentioned the use of complementary label learning for estimating the distribution of negative coarse labels. Furthermore, it is claimed in Section 3.1 that the proposed method is also applicable to cases involving only positive or negative coarse labels. Does the model work with only negative coarse labels? I am expecting to see the corresponding experiments/ablation studies to support the claims (only uses negative coarse labels).
- One of the basic/main assumptions in this paper is that: Given the input image, the authors assume that the provided coarse annotations are generated statistically independently across different samples and over different pixels. However, in practice, I am afraid this is not always true. The spatial relationship exists within neighboring/adjacent pixels and their corresponding coarse labels. Any thoughts regarding this?
- In Table 1, the proposed method also compares with two corse annotation-based baselines. While there are a few more recently proposed methods [1,2], just list a few. Also, in Table 2 of [2], the reported numbers on Cityscape seem to outperform the numbers reported in this paper. Would you please include such baselines as well for a fair comparison?

[1] Saha, Oindrila, Zezhou Cheng, and Subhransu Maji. "Improving few-shot part segmentation using coarse supervision." European Conference on Computer Vision. Cham: Springer Nature Switzerland, 2022.

[2] Das, Anurag, et al. "Urban Scene Semantic Segmentation with Low-Cost Coarse Annotation." Proceedings of the IEEE/CVF Winter Conference on Applications of Computer Vision. 2023.

**Questions:**

See the Weakness section.

---

> ### Author Response · Authors · 2023-11-12
>
> Q1: The reviewer is correct in pointing out that there are no specific experiments or ablation studies presented in the paper to support this claim for the case of using only negative coarse labels. In this work, there are limitations of using only negative coarse labels. Our model leverages the information from negative coarse labels to improve the estimation of true segmentation label distributions, the absence of positive coarse labels may result in a lack of guidance for the model in identifying the specific regions to be segmented, which could lead to an increased number of false positives.
>
> Q2: we agree with reviewer that independence assumption between pixels in annotation might limit the performance. We note, however, that the annotations are assumed to be only conditionally independent between pixels given the input image, and thus the model can still capture some correlations in the output segmentation labels that are explained in the input image. We additionally note that such independence assumption is typically made in most of the deep learning based segmentation methods, and thus a posprocessing method such as Gaussian CRF is commonly used to capture the missed correlations. We believe the same problem applies to the annotation modelling---we note this limitation and mention such correlation modelling as future work in the discussion.
>
> Q3: Thanks for introducing more recent coarse annotation-based baselines for comparison. As we see from [1] and [2], they use the coarse annotation provided by Cityscapes dataset for evaluation, however our work chose 30% of the provided coarse annotation for evaluation, which is more challenging compared to [1] and [2]. This is also the reason why our model’s performance is worse than [2]. We will clarify this in the experiment section of the paper.

---

> > ### Comment · Reviewer_fJPX · 2023-11-23
> >
> > Thank you for your detailed response and it solved most of my concerns. I will raise my rating to 6 to reflect this.

---

### Meta-Review · Area_Chair_MGcM · 2023-12-06

**Metareview:**

This paper proposes a method for learning semantic segmentation from coarsely annotated labels. This is an important problem in the sense that annotating segmentations is time consuming and expensive. It remains an important problem even in the face of new segmentation paradigms such as SegmentAnything, as these do not work out of the box on challenging medical imaging segmentation tasks -- certainly not on the retinal vessel segmentation included in this paper.

That being said, the authors' approach has limited novelty, as pointed out by the reviewers, and even more importantly, their validation is performed on unrealistic tasks where the data does not broadly represent medical imaging, and the coarse segmentations are not real, but synthesized using morphological operations. This is not necessary -- an ICLR'23 paper (https://openreview.net/forum?id=wZ2SVhOTzBX) curates biomedical imaging datasets (and solutions) for this precise task.

The latter criticism in particular brings the paper below the acceptance limit for ICLR'24, and I cannot recommend acceptance. I encourage, however, the authors to utilize the reviews to improve the paper and resubmit to a good conference -- they are indeed working on an important problem.

**Justification For Why Not Higher Score:**

The validation is not sufficiently realistic and well described. These concerns are sufficiently serious that I cannot recommend acceptance.

**Justification For Why Not Lower Score:**

N/A

---

### Decision · Program_Chairs · 2024-01-16

Reject